# Practical Lessons on Antimicrobial Therapy for Critically Ill Patients

**DOI:** 10.3390/antibiotics13020162

**Published:** 2024-02-06

**Authors:** Rachael Cusack, Elizabeth Little, Ignacio Martin-Loeches

**Affiliations:** 1Department of Intensive Care Medicine, Multidisciplinary Intensive Care Research Organization (MICRO), St James’ Hospital, D08 NHY1 Dublin, Ireland; rqcusack@gmail.com (R.C.); elizabethlittle@rcsi.ie (E.L.); 2Hospital Clinic, Universitat de Barcelona, IDIBAPS, CIBERES, 08180 Barcelona, Spain

**Keywords:** intensive care, hospital mortality, antibiotics, sepsis, shock, MDR, VAP

## Abstract

Sepsis stands as a formidable global health challenge, with persistently elevated mortality rates in recent decades. Each year, sepsis not only contributes to heightened morbidity but also imposes substantial healthcare costs on survivors. This narrative review aims to highlight the targeted measures that can be instituted to alleviate the incidence and impact of sepsis in intensive care. Here we discuss measures to reduce nosocomial infections and the prevention of equipment and patient colonisation by resilient pathogens. The overarching global crisis of bacterial resistance to newly developed antimicrobial agents intensifies the imperative for antimicrobial stewardship and de-escalation. This urgency has been accentuated in recent years, notably during the COVID-19 pandemic, as high-dose steroids and opportunistic infections presented escalating challenges. Ongoing research into airway colonisation’s role in influencing disease outcomes among critically ill patients underscores the importance of tailoring treatments to disease endotypes within heterogeneous populations, which are important lessons for intensivists in training. Looking ahead, the significance of novel antimicrobial delivery systems and drug monitoring is poised to increase. This narrative review delves into the multifaceted barriers and facilitators inherent in effectively treating critically ill patients vulnerable to nosocomial infections. The future trajectory of intensive care medicine hinges on the meticulous implementation of vigilant stewardship programs, robust infection control measures, and the continued exploration of innovative and efficient technological solutions within this demanding healthcare landscape.

## 1. Introduction

Antimicrobial therapy in the intensive care unit (ICU) is ubiquitous, with approximately 71% of patients receiving antibiotics [1]. Many antimicrobial drugs used can cause harm to critically ill patients, who often have multiple organ impairments [2,3,4,5,6,7]. Balancing potential damage from antimicrobial medications against appropriate source control and treatment of infection presents a challenge; in this narrative review, we aim to outline considerations and barriers to appropriate antimicrobial therapy for the sickest patients in hospitals. Nosocomial infection, and especially hospital and healthcare-associated pneumonia (HAP), are common causes of morbimortality in ICUs around the world [8]. Immuno-protective mechanisms of the respiratory system are impaired by acute illness and invasive interventions in the ICU, leaving patients vulnerable. Patients can develop ventilator-associated pneumonia (VAP), which is associated with unique phenotypes of sepsis and respiratory failure that are then superimposed on their underlying illness. The mortality rate of patients with VAP was 36% internationally according to a 2020 report [8].

Increasing antimicrobial resistance patterns create further difficulties, and novel methods to rationalise and de-escalate antimicrobial therapy are necessary for modern intensive care therapy [9]. Recent developments in antimicrobial treatment include introducing novel drug delivery systems and recognising the importance of stewardship programs to curb the growth of resistant pathogens [10,11,12]. For example, pharmacokinetics in critically ill patients are altered by mechanical ventilation, renal replacement therapy, and pathological processes [13,14,15]. The pharmacodynamics of antimicrobial drugs can also be changed [16]. Artificial nutrition, sepsis, and antimicrobial treatment can lead to dysbiosis and microbiome alteration, which are linked with increased morbidity and mortality [17,18,19].

Colonisation of the respiratory tract is recognised as a predisposing factor to respiratory infections in critically ill patients. There is increasing evidence for bundles that reduce the impact of acute illness on the protective mechanisms of the respiratory system and for respiratory drug delivery devices. Microbiology advice is regional and fluctuating, constantly changing with the migration of populations and pathogens [20,21,22]. This can introduce doubt in practitioners and is a constant challenge to staying abreast of developments and guidelines. 

This text aims to outline lessons on antimicrobial therapy in intensive care that can be applied daily by practitioners. Understanding the mechanisms of microbial mutations and mechanisms of action of classes of antimicrobials is necessary for a successful career in intensive care. We also discuss potential future directions and research areas that will revolutionise the future. 

## 2. Methodology

We meticulously crafted a comprehensive search strategy for the manuscript, focusing on infections in critical care, utilising a combination of keywords such as ‘critical care’, ‘infection’, and ‘antimicrobial therapy’, and employing Boolean operators to refine the search parameters, ensuring a thorough exploration of relevant literature in databases such as PubMed and MEDLINE. 

## 3. The Role of Colonisation in Critically Ill Patients

Colonisation with bacteria is common. Colonisation refers to the existence and survival of microbes on surfaces, internal (gastrointestinal, respiratory, or genitourinary) or external (skin), that do not produce disease in the host. Healthcare workers and inanimate surfaces in the ICU can be colonised with various drug-resistant pathogens that have been shown to cause nosocomial infection [23,24,25]. Carriers of resistant pathogens are at increased risk of developing infection with resistant microbes [25]. This has led to the development of solid infection prevention and control (IPC) programs in the last 50 years [26]. Many patients in intensive care may have had a prolonged hospital stay or multiple hospitalisations and are often colonised with resistant pathogens [27,28,29]. Commensal bacteria are essential in protecting the host from these pathogens [30]. When this balance is disturbed following hospitalisation or treatment of colonising bacteria with antibiotics, it can contribute to the development of infection, resistance, and patient harm. Immunosuppression, either pharmacologically or pathologically, or alteration of the microbiome can cause commensal bacteria to give way to infection. 

The problem of colonisation presented itself in particular during the SARS-CoV2 pandemic, when large numbers of patients were treated with high-dose steroids and immunological agents [31,32]. Normal colonising bacteria can develop into virulent infections in chronically immunosuppressed patients such as transplant recipients or cancer patients. Primary candida pneumonia is rare in the healthy population, but in those that are treated with immune-modulating drugs, secondary *Candida* pneumonia can be severe [33]. Invasive candidiasis manifests in 1–8% of hospitalised patients overall. However, its incidence rises to approximately 10% among individuals in ICUs, constituting up to 15% of all nosocomial infections in this setting [34]. However, treating *Candida* spp. grown from sputum in ICU is associated with increased morbidity and mortality [35]. Patients who receive antifungal treatment for *Candida* isolated from mucosa or respiratory samples, without displaying signs of infection, are at increased risk of worse outcomes in ICU and hospital. *Candida* isolation on admission to critical care is associated with increased severity, but treating with antifungals does not improve outcomes, as has been seen by observation [36] and in a randomised controlled trial [37], respectively.

The ICU also proved to be a source of infection with drug-resistant pathogens in the COVID-19 pandemic. Shortages of personal protective equipment and staffing shortages result in reduced time between patients and decreased usual attention to inter-patient hand hygiene or PPE changes [38]. Many units were advised to use alcohol gel on latex gloves to conserve PPE instead of applying new gloves and washing hands [39]. Antimicrobial stewardship activities and everyday hygiene audit practices or resistant microbe tracing programs were foregone in the face of the overwhelming COVID-19 spread. This is particularly concerning, as not only are patients often carrying multi-drug-resistant pathogens, but these pathogens can colonise the ICU environment [40].

The gut and lungs are essential reservoirs of commensal protective pathogens. The use of invasive devices in intensive care, such as central venous access devices, endotracheal tubes, and urinary catheters, can also act as reservoirs of infection [41,42,43]. In the ICU, gut-to-lung translocation of commensal bacteria can cause severe infection [44,45,46]. Microbes that reside harmlessly in the intestine lead to severe lung infection. Catheters can also harbour harmful pathogens, although positive urinary cultures do not always indicate infection requiring antimicrobial treatment. Specific commensals, such as *Staphylococcus aureus*, which can inhabit a patient’s skin or nasopharynx gain virulence when invasive devices are introduced [23]. *Streptococcus pneumoniae*, a common lung commensal, can cause deadly infection in elderly or young patients with immune downregulation following hospitalisation or illness [47,48].

## 4. Antimicrobial Stewardship and the Dilemma of Broad vs. De-Escalation in ICU 

As the prevalence of multi-drug-resistant pathogens continues to escalate, the implementation of the Antimicrobial Stewardship Strategy (AMS) emerges as a critical line of defense against this pressing issue. Antimicrobial stewardship encompasses a comprehensive suite of interventions strategically designed to curtail inappropriate antimicrobial prescribing, mitigate costs, and combat the escalating problem of antimicrobial resistance [49,50]. In recent years, the use of antimicrobials in ICUs has been notably high, with an extended prevalence study indicating that 61% of medical and surgical ICU patients receive antibiotics [51]. Given this substantial reliance on antimicrobials within the ICU setting, it becomes evident that the ICU serves as a natural and crucial target for effective AMS interventions.

Within the realm of ICU care, where critically ill patients are highly susceptible to infections, the judicious use of antimicrobials takes on paramount significance. The multifaceted nature of AMS interventions within the ICU involves not only curbing inappropriate prescribing but also addressing cost implications and actively combating the emergence of antimicrobial resistance. By targeting the ICU setting, AMS initiatives aim to optimise the use of antimicrobial agents, ensuring that patients receive the most effective treatments while simultaneously safeguarding the efficacy of existing drugs.

AMS has traditionally been associated with the concept of restricting antibiotics in clinical practice. While limiting antibiotic use is indeed one of its fundamental principles, the overarching aim of AMS extends beyond mere restriction to emphasise the importance of the appropriate and judicious use of antibiotics. In essence, AMS seeks to ensure that antibiotics are prescribed discerningly, avoiding unnecessary administration to patients who do not require them while ensuring timely and effective treatment for those with infections.

While the rationale behind AMS may seem straightforward, its application is inherently complex in the dynamic landscape of daily clinical practice. Healthcare providers face a delicate balance, navigating the intricacies of patient presentations, diagnostic uncertainties, and the evolving landscape of antimicrobial resistance. AMS strategies involve a nuanced understanding of microbial infections, tailoring interventions to individual patient needs, and considering the broader implications of antimicrobial use on both the patient and public health. As such, AMS serves as a comprehensive framework that goes beyond restriction, fostering a culture of responsible and patient-centered antibiotic utilisation in the complex and dynamic realm of clinical care. One of the main difficulties faced by practitioners on a daily basis is the problem of de-escalation. Frequently susceptibilities may be found for infective microbes but patients continue to be extremely sick. The dilemma presents then itself: is it appropriate to wean antibiotic cover in a critically ill patient? Often we feel we cannot, despite apparently reassuring microbiological tests. 

Reducing mortality is the central objective of addressing medical interventions employing appropriate tools. Laboratory testing can play a pivotal role in various aspects of AMS, particularly in facilitating de-escalation and optimizing the duration of interventions [52,53]. 

## 5. Which Resistance Mechanisms Should an Intensivist Know?

Antimicrobial resistance (AMR) is at the top of the global public health priorities list, per recent World Health Organisation (WHO) documents. It is estimated that only bacterial AMR (excluding fungal AMR) has been reported to be directly responsible for 1.27 million global deaths and contributed to 4.95 million deaths in 2019 [54,55]. It is essential to understand the mechanisms of resistance when choosing alternative therapy [56]. Most important is to identify at-risk patients who may be colonised with resistant organisms so that they can be targeted with appropriate therapy, while reducing exposure of the general population to second or third line antimicrobial therapy. Resistance mechanisms can also guide potential future drug targets. In Figure 1, we displayed the most common mechanisms of bacterial resistance. Antimicrobial resistance mechanisms are broadly categorised into four main types, each representing distinct strategies employed by microorganisms to withstand the effects of drugs. Firstly, microorganisms may limit the uptake of a drug, restricting its entry into the cell. Second, they may modify the drug target, altering the specific site within the microorganism that the drug aims to interact with. Another mechanism involves the inactivation of the drug, rendering it ineffective in its intended function. Lastly, microorganisms may employ active drug efflux, expelling the drug from the cell before it can exert its antimicrobial effects. This intricate web of resistance mechanisms highlights the adaptability and resilience of microorganisms in the face of antimicrobial agents. The first category underscores the importance of preventing drug entry, while the second emphasises the need for novel drug targets that resist modification. Additionally, the inactivation mechanism emphasises the ongoing challenge of developing drugs that remain effective even when faced with microorganisms’ attempts at neutralisation. Finally, the active drug efflux mechanism underscores the significance of designing drugs that can withstand expulsion from microbial cells, ensuring their sustained efficacy. Understanding and addressing these diverse resistance mechanisms are imperative in the ongoing battle against antimicrobial resistance. It necessitates a multifaceted approach involving the development of innovative drugs, enhanced surveillance systems, and global collaborative efforts to preserve the effectiveness of existing antimicrobial agents and pave the way for a sustainable future in infectious disease management [56].

As a result, we have determined the most common resistance mechanism that a physician working in the ICU should know. We mainly acknowledge bacterial mechanisms as previous epidemiological studies’ most common pathogens isolated in critical care settings [1]. This article explores fundamental aspects of antibacterial resistance, encompassing mechanisms and transmission modes, and delves into the management considerations for key drug-resistant pathogens encountered in the ICU. Gaining a deeper understanding of these mechanisms is anticipated to result in improved treatment choices for infectious diseases and the creation of antimicrobial drugs capable of resisting the microorganisms’ attempts to develop resistance.

### 5.1. Mechanisms of Antimicrobial Resistance

The early emergence of β lactamase, which appeared soon after the first clinical use of penicillin, is evidence that bacteria encounter antimicrobial compounds in their natural evolutionary environment. This could explain why bacteria have developed resistance mechanisms that can appear so swiftly in the clinical setting. Bacteria with corresponding antibacterial compounds require methods of overpowering the lethal effects of antimicrobial particles. They have evolved to do this in several ways, such as the expression of diverse particle targets, the creation of degrading enzymatic processes, and the expulsion of antibiotic complexes. Bacteria are capable of sharing resistance mechanisms, and since many of the antibiotics created in recent decades are manipulated versions of older antibiotics, bacteria are able mutations to disseminate resistant gene mutations quickly.

### 5.2. Gram Positive Resistance

β-lactam antibiotics inhibit bacteria such as *Staphylococcus aureus* through cell wall formation interruption. This first β-lactamase is known as penicillinase [57]. Peptidoglycans in the cell wall are vital for the survival and functioning of bacteria and so they cannot evolve to survive without them, making this the frontline in the war against bacterial resistance. Future generations of *S. aureus* developed novel penicillin-binding proteins (PBP) with which methicillin and penicillin-derived β-lactam antibiotics could not interact, conferring resistance [58]. Fifth-generation cephalosporin ceftaroline was developed to bind to this altered PBP, called PBP2a [59]. Methicillin-resistant *S. aureus* was initially regarded as a healthcare-associated bacterium but has become more prevalent in the community, emphasising how resistance moves among generations of bacteria [60]. 

Vancomycin is an antibiotic that prevents elongation of the PBP chain, interrupting the cell wall. However, since 1997, resistance to vancomycin has also emerged [61]. Bacteria can downregulate genes and thicken their cell wall in response to prolonged exposure to vancomycin, resisting its effects [62,63,64]. *S. aureus* resistant to vancomycin inherited the *vanA* gene from a strain of vancomycin-resistant enterococcus. 

Vancomycin resistance first appeared in *Enterococcus faecium*, over 30 years ago [65]. This is another example of altered protein binding as bacteria alter the d-alanyl-d-alanine terminal peptide to d-alanyl-d-lactate, a mutation coded by the genes *vanA* and *vanB* [66]. Treatment of these resistant strains led to developing linezolid and daptomycin, an oxazolidinone and a linopeptide, respectively. However, resistance to these antimicrobials has also developed [67,68]. By mutating the 23S rRNA as a result of exposure to linezolid, bacteria can prevent linezolid binding to its 50S ribosome. Daptomycin halts cell wall synthesis by binding to phosphatidylglycerol, depolarising the bacteria. Thankfully, daptomycin resistance remains low, though bacterial cell walls continue to adapt and evolve. 

### 5.3. Gram-Negative Resistance 

In 2019, more than 250,000 deaths each were attributed to *Escherichia coli*, *Klebsiella pneumoniae*, *Acinetobacter*, and *Pseudomonas aeruginosa.* Carbapenem, cephalosporin, and fluoroquinlone resistant strains of each of these pathogens were responsible for over 50,000 deaths each globally [69]. A study in hemodialysis patients in the United States found that nearly a third of patients showed evidence of colonisation with Gram-negative bacteria that carried resistance to 50% of antimicrobial agents in the study. A further fifth of patients studied acquired one of these pathogens in the following six months [70]. 

The *Enterobacteriaceae* family includes various bacteria such as *Escherichia coli*, *Klebsiella* spp., and *Enterobacter* spp., all playing significant roles in infections associated with both community and healthcare settings. However, there is an increasing prevalence of MDR *Enterobacteriaceae* in community environments [71,72,73]. 

Extended-spectrum β-lactamases (ESBLs) cleave the β-lactam ring and are transferred in plasmids. The first β-lactamases were TEM-1, TEM-2, and SHV-1, and point mutations in these enzymes led to ESBL emergence. These enzymes are resistant to β-lactams, cephalosporins and oxyimino-β-lactams(ceftriaxone, ceftazidime and cefotaxime). Most ESBLs are still susceptible to β-lactam-β-lactamase inhibitor combinations such as pipercillin-tazobactam. 

In contrast, bacteria carrying AmpC β-lactamase mutations are not inhibited by β-lactamase inhibitors [74]. The leading producers of AmpC enzymes are pathogens referred to as ‘SPICE’ (*Serratia*, *Providencia*, indole-positive *Proteus*, *Citrobacter*, and *Enterobacter* spp.). This resistance mechanism is inducible upon exposure to a β-lactamase inhibitor drug, and seemingly susceptible pathogens can fail to respond to ‘appropriate’ treatment. AmpC genes are now commonly transmitted in plasmids, though they are usually chromosomal [74]. This increases their transmission and prevalence. 

The family of β-lactamases is increased by the carbapenemases that hydrolyse β-lactam antimicrobials. Cabapenamases are resistant to β-lactamase inhibitors but also carry several genes granting resistance to other drug classes and can confer multi-drug resistance between generations. Class B metallo-β-lactamases (MBLs) in the Ambler classification (Figure 2) are a large group of worrying resistance genes now seen worldwide, first identified in Japan [75]. The Oxacillinase-48 carbapenemase (Oxa-48) is becoming more globally recognised, once more commonly seen in North Africa and the Middle East [76]. Although geographic resistance patterns were more specific when they first emerged, with international travel and migration patterns, MDR bacteria carrying carbapenemases are becoming widespread globally [77,78]. Options to treat carbapenemase-producing enzyme bacteria are narrow. In 2017, the European Centre for Disease Prevention and Control reported that 7.2% of invasive *K. pneumoniae* in 30 European countries carried carbapenem resistance [79]. In 2021, 8 out of 45 countries reporting to the ECDC had resistance rates above 50%, 15 countries had resistance rates of 25%, and only 14 countries reported resistance below 1% [80]. The International Nosocomial Infection Control Consortium (INICC) published a global report of ICUs from developing countries 2012 to 2017 stating that 37% of *K. pneumoniae* found in blood culture samples were carbapenem-resistant [8]. Increased porin production and drug efflux pumps also convey resistance to carbapenems.

### 5.4. Alternative Resistance Mechanisms

Changes in the DNA gyrases *gyrA* and *parC* confers resistance against quinolone antibiotics, which is becoming increasingly common. Quinolone resistance also occurs due to mutations in chromosomally encoded porin channels and efflux pumps [81]. 

Aminoglycoside-modifying enzymes can also confer resistance. These resistance genes are often transmitted on plasmids that also contain KPCs or ESBLS [82,83]. However, resistance to aminoglycosides tends not to develop during treatment for an aminoglycoside susceptible pathogen. 

*Pseudomonas aeruginosa* is often associated with VAP and ICU and has many resistance mechanisms, including AmpC β-lactamases, ESBLs, MBLs, downregulation of porins and upregulation of efflux pumps [84,85]. *P. aeruginosa* resistance is responsible for increased morbimortality as well as healthcare costs in ICU settings [8,86,87]. In 2014, *P. aeruginosa* was the 6th most common bacteria in surgery- and device-associated infections, and pseudomonal biofilms in the ICU are a unique problem in this setting [84,88]. In 2021, a third of isolates were resistant to at least one anti-pseudomonal antimicrobial agent; however, 5% of isolates were resistant to four or five antimicrobial groups [80]. 

*Acinetobacer baumanii* resistance is a growing problem globally and especially in ICUs, where the bacteria have been responsible for rapidly developing resistance and causing outbreaks [89,90,91]. OXA carbapenemases are increasingly found in *A. baumanii* species. The INICC reported that 92% of *Acinetobacter* VAP was resistant to carbapenems and 66% of *Acinetobacter* species in Europe were resistant to fluoroquinolones, aminoglycosides, and carbapenems in 2021 [8,80].

## 6. Antibiotic Dosing

In a worldwide point prevalence study of ICUs in 2015, 54% of patients were being treated for suspected or proven infection, with in-hospital mortality of 30% [51]. Dosing of antimicrobials and applying knowledge of PK/PD principles is crucial to optimising bacterial killing at the infection site, reducing antimicrobial resistance, minimizing adverse drug reactions and toxicity, and preserving the lifespan of available antimicrobials [92,93].

### 6.1. Altered Physiology in Critical Care Patients

Intensive care patients with sepsis exhibit dynamic pathophysiological changes that can alter antimicrobial concentrations. The sepsis inflammatory response leads to significant fluid shifts into the interstitial space, an initial high cardiac output, and hypoalbuminaemia, leading to increased volume of distribution (Vd); less antimicrobial is available in plasma and, therefore, at the site of infection [92,94]. For other patients, organ dysfunction with renal or hepatic impairment may require reduced dosing to ensure therapeutic but non-toxic levels [95]. Acute kidney injury requiring CRRT occurs in 15% of septic patients and increases mortality by 50%. Alteration of antimicrobial dosing in renal replacement therapy (RRT) extracorporeal organ supports should be individualised [96,97].

### 6.2. Potential Drug Mutant Concentrations

Antimicrobial choice and resistance have traditionally been derived from laboratory minimum inhibitory concentration (MIC) values [98]. Having antimicrobial dosing instead based on the mutant prevention concentration (MPC) has been proposed to potentially prevent the growth of first-step resistant mutants, as ICU patients with sepsis have an initial high bacterial burden with an increased likelihood of first-step resistance mutants, requiring an increase in antimicrobial dosing [99,100]. This is balanced with the risk of drug toxicity, a dose modulation strategy with early, higher antimicrobial doses (selected based on PK and infection characteristics) followed by dose reduction when hemodynamics improve, to optimise antimicrobial dosing [101].

### 6.3. Increased Creatinine Clearance

Augmented renal clearance (ARC), defined as a creatinine clearance > 130 mL/min, may be seen in 20–65% of intensive care patients from a hyperdynamic state with increased cardiac output and enhanced renal blood flow. It can significantly increase the clearance of aminoglycosides, β-lactams, and glycopeptides, leading to subtherapeutic antimicrobial concentrations [102,103].

### 6.4. Nebulisation 

Nebulised antimicrobials used as adjunctive therapy in VAP may facilitate antimicrobial concentrations well above MIC, enhancing lung parenchymal penetration, reducing the rate of MDR pathogens, and lowering systemic toxicity, provided appropriate ventilatory settings, doses, and devices are used [93]. Three RCTs (IASIS, INHALE and VAPORISE) were studied for their use in β-lactam- and fluoroquinolone-sensitive GNB VAP did not show mortality benefit or improvement in clinical cure rate. The efficacy of these studies may be compromised due to lower daily doses of aminoglycosides and polymyxins and the sensitivity of GNB pathogens [104,105,106,107].

Nebulised antimicrobials are currently recommended by IDSA/ATS guidelines in GNB, pan-, and extensively drug-resistant (PDR, XDR) VAP where there is poor lung penetration or systemic toxicity from intravenous treatment (e.g., aminoglycosides and polymyxins) [108]. Future research on the most appropriate nebulisation device and if antimicrobial nebulisation in drug-resistant GNB should be a substitution or adjunct to IV treatment is welcomed [109]. 

### 6.5. Prolonged Infusions vs. Intermittent Administration

The most recent Surviving Sepsis Guideline (2021) recommends a prolonged (extended or continuous) β-lactam infusion. However, evidence for this is of moderate quality [100]. The postulated benefit is from a higher cumulative percentage of time achieved above MIC (fT>MIC), which may lead to more rapid bacterial killing and faster clinical improvement. The Defining Antibiotic Levels in Intensive Care Unit Patients (DALI) study across 68 ICUs assessed whether β-lactam dosing (prolonged and intermittent) achieves drug concentrations correlating with maximal activity and if this leads to favourable clinical outcomes [110]. The results showed wide variation in β-lactam plasma concentration (20% patients not achieving a target of >50%fT>MIC), making 1 in 3 less likely to have favourable clinical outcomes. Patients receiving prolonged infusion therapy were more likely to achieve the pharmacodynamic target than those receiving intermittent therapy (93% vs. 80%, respectively). Several systematic reviews and meta-analyses have aimed to synthesise the current mortality data on prolonged vs. intermittent β lactam infusions in sepsis and severe sepsis [110,111,112,113]. Kondo et al. and the more recent MERCY trial did not infer an in-hospital mortality or 90-day mortality benefit, respectively, with prolonged infusions, in contrast to Vadarkas et al. [108,110,113]. Others specifically addressed short-term (30-day) mortality and clinical cure and did infer some benefit [113,114]. All these systematic reviews were underpowered, and there were variable definitions of sepsis and clinical cure and considerable performance bias. We look forward to high-quality evidence from the recently completed multicentre Beta-Lactam InfusioN Group Study (BLING III) of over 7000 ICU patients [115].

## 7. Therapeutic Drug Monitoring and Dosing Software

As early effective antimicrobial therapy in sepsis improves mortality and dose optimisation, taking into account the MIC of the pathogen and the individual’s dynamic PK throughout the clinical course (to refine dosing) is paramount in achieving antibacterial killing and reducing toxicity (Table 1) [100]. Traditional dosing nomograms may not apply to an ICU patient or the pathogen’s specific MIC [116,117]. Therapeutic drug monitoring (TDM) is when a serum sample is obtained at a defined time post antimicrobial administration, with a short turnaround time for the result. Compared to empiric dosing, it has been shown to increase the proportion of patients achieving PK-PD targets [30], and its utilisation has expanded. It is recommended for β-lactams, aminoglycosides, and glycopeptides [93,118,119]. In a recent international survey, 82% of respondents utilised TDM for aminoglycoside and 90% for vancomycin [120].

The dosing software model, description, and application are given below [121]:

**Table 1 antibiotics-13-00162-t001:** Dosing software model, description and application.

Dosing Software Model	Therapeutic Drug Monitoring Required	Pharmacokinetic Models	Pathogen-Specific MIC Targets	Application
Linear regression [122]	Yes 2 plasma concentrations at different time points	No	No	Drug clearance rate and subsequent dose recommendation
Population PK model [123]	Yes at least 1 plasma concentration	Yes Pre-specified PK-PD drug targets give initial dosing recommendation	No	Dosing range recommendation
Bayesian forecasting	Yes 1 plasma concentration at least	Yes Pre-specified PK-PD drug targets give initial dosing recommendation	Yes Can incorporate MIC targets	Generates most appropriate antimicrobial dosing required based upon PK-PD models and TDM and MIC variation
Artificial intelligence	Yes Improved accuracy with 2 plasma concentration samples	Yes Reinforcement learning from large databases	Yes Can incorporate MIC specific targets	Individualised for patient to optimise favourable outcome from re-inforcement learning by adjusting dosing, addressing and compensating for drug interactions

Whilst there is evidence for artificial intelligence (AI) in the recognition and treatment of sepsis in ICU patients, its use in antimicrobial dosing is not yet established [124,125].

### 7.1. Dosing Software

Dosing software in critical care has been implemented recently with specialised algorithms to help determine and administer appropriate medication doses for critically ill patients. These software solutions are designed to enhance accuracy, efficiency, and safety in medication management within the complex and dynamic environment of critical care settings. Implementing dosing software in ICU can improve patient outcomes, reduce medication errors, and increase healthcare delivery efficiency. It is essential for healthcare institutions to carefully evaluate and integrate such software into their existing systems while ensuring proper training for healthcare professionals. Regular updates and maintenance are also crucial to keep the software aligned with the latest medical knowledge and standards. Key features and functionalities of dosing software in critical care may include patient-specific data integration, clinical decision support, pharmacokinetic and pharmacodynamic models, continuous monitoring and adjustment, compatibility with infusion devices, drug interaction alerts, and automated compliance with protocols and guidelines, with a user-friendly interface that guarantees security and privacy with documentation and reporting tools.

### 7.2. Future Directions

Potential developments in antimicrobial treatment in ICU are really encouraging. The integration of AI stands out as a transformative force in this domain. AI has the capacity to revolutionise antimicrobial stewardship by leveraging computer programs to meticulously track resistance patterns and dissemination trends within hospitals and broader regions. This data-driven approach allows for real-time monitoring, enhancing the precision of interventions and optimizing treatment strategies.

Furthermore, there is a growing emphasis on the importance of phenotyping and endotyping patients within the ICU setting. By characterizing individuals based on their unique biological and clinical profiles, healthcare professionals can tailor antimicrobial interventions to specific patient needs. This personalised medicine approach holds promise in not only improving treatment efficacy but also mitigating the risk of adverse effects and promoting better overall patient outcomes.

Additionally, the advent of point-of-care sensitivity testing represents a paradigm shift in antimicrobial management. This technology enables rapid assessment of microbial susceptibility directly at the patient’s bedside, facilitating timely adjustments to treatment regimens. The immediacy of results empowers healthcare providers to make informed decisions, optimizing the selection of antimicrobial agents and minimizing the potential for resistance development.

As these innovative approaches gain traction, the future of antimicrobial treatment in the ICU is envisioned as a dynamic and adaptive landscape. Integrating cutting-edge technologies, embracing personalised medicine principles, and fostering a proactive stance through point-of-care testing collectively hold the potential to reshape the way infections are managed in critical care settings. The ongoing pursuit of these advancements underscores the commitment to enhancing patient care, reducing mortality rates, and effectively addressing the challenges posed by antimicrobial resistance in the ICU.

## 8. Conclusions

Within the dynamic environment of the ICU, the meticulous management of antibiotic resistance and the vigilant administration and monitoring of antibiotics stand as pivotal elements in providing optimal patient care. The ICU population, marked by the severity of underlying illnesses and the intensive treatments they receive, are notably vulnerable to a spectrum of nosocomial infections. This susceptibility is further heightened by the potential colonisation of equipment and alterations in the patient’s pathogens, both playing critical roles in the development of infections.

Addressing this multifaceted challenge, the implementation of antimicrobial stewardship programs and infection prevention bundles has emerged as a crucial strategy. These initiatives have demonstrated efficacy in mitigating the escalating threat posed by rapidly spreading bacteria, many of which exhibit resistance to our most potent antibiotics. The urgency for innovative solutions has catalyzed ongoing developments in novel antimicrobials and cutting-edge technologies, with a specific focus on enhancing diagnostic capabilities and refining drug monitoring processes. These advancements, anticipated on the near horizon, hold the potential to reshape the entire landscape of critical care antimicrobial treatment.

The evolution of precision medicine principles within the ICU is becoming increasingly apparent, as tailored approaches to address infections gain prominence. The commitment to remaining at the forefront of antimicrobial research is imperative, given the ever-evolving challenges within the ICU setting. By embracing emerging technologies, optimizing diagnostic accuracy, and continually refining therapeutic strategies, the ICU can fortify its capacity to combat infections effectively and enhance patient outcomes. As the field progresses, the integration of innovative antimicrobial solutions promises to provide a more nuanced and sophisticated framework for addressing the diverse and complex infection challenges encountered in the critical care setting.

## Figures and Tables

**Figure 1 antibiotics-13-00162-f001:**
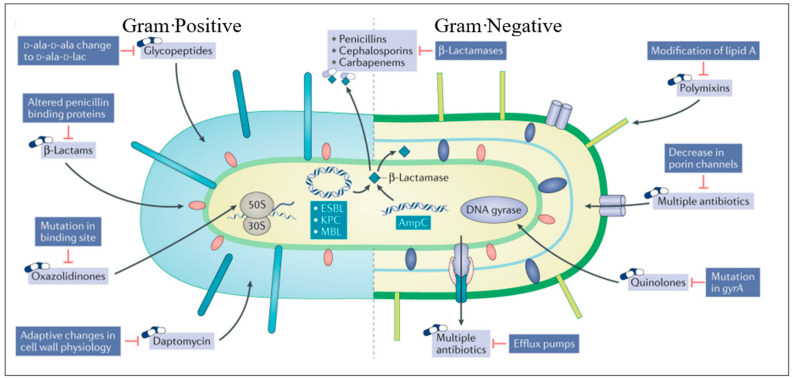
Sites of action of commonly used antimicrobials and the resistance mechanisms that adapted to them (adapted from [56]). This figure illustrates the sites of action targeted by frequently employed antimicrobials and the corresponding resistance mechanisms that have emerged against them.

**Figure 2 antibiotics-13-00162-f002:**
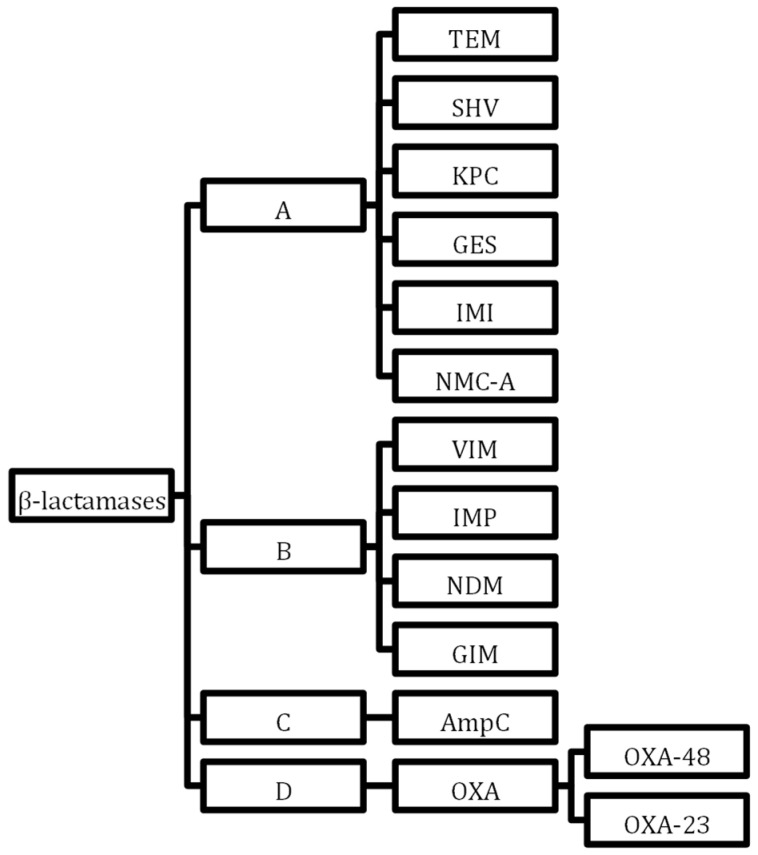
Ambler classification of β-lactamases. In accordance with the Ambler classification system, β-lactamases undergo classification into four distinct categories: Class A, Class B, Class C, and Class D. This categorisation is determined by particular motifs inherent within the primary sequences constituting the protein molecules. Notably, Class A, Class C, and Class D β-lactamases employ a serine residue within their enzyme active sites to catalyze reactions, whereas Class B β-lactamases rely on zinc ions to facilitate their catalytic activity.

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
