# Peer review of "Practical Lessons on Antimicrobial Therapy for Critically Ill Patients"

_antibiotics, 2024, doi:10.3390/antibiotics13020162_

Round 1

Reviewer 1 Report

Comments and Suggestions for Authors

The article “Practical lessons on antimicrobial therapy for critically ill patients” is a narrative review focusing on an important, and widely discussed problem. The scope of the review is too wide and the article promises to discuss “the barriers and facilitators to effectively treating critically ill patients at risk of nosocomial infection”.

Specific comments:

1.     The title does not properly describe the content of the article. The manuscript focuses on general information about antimicrobial resistance, resistance mechanisms, and even on microbiome.

2.     Bacteria names are misspelled in the whole document. For example, on Line 53 „Clostridium diffficile” is mentioned. The name changed in 2016, and the correct one is „Clostridioides difficile”

3.     Line 53 – although the microbiome is an interesting topic, the line and later on, a whole section about microbiome does not fit in a narrative review titled „Practical lessons on antimicrobial therapy for critically ill patients”

4.     “Colonisation” vs “Infection” section: the whole section focuses on colonization, while the infection is overlooked;

5.     „The role of the microbiome” section – does not fit in the scope of the review

6.     „Antimicrobial stewardship....” section – contains general, well-known information about the importance of antimicrobial stewardship

7.     „Which resistance mechanisms should an intensivist know” section – the title is catchy but the question was not answered in the text. The section describes a few resistance mechanisms, The information is not complete and more importantly, it lacks novelty. Any clinician should have at least a basic understanding of the resistance mechanisms, especially if they have epidemiological value.

Overall, the review does not have a well-defined scope. The importance to the readership is poor, and the search strategy is not presented.  

Comments on the Quality of English Language

Please review the manuscript and use the proper scientific nomenclature for bacteria/fungi names.

Author Response

The article “Practical lessons on antimicrobial therapy for critically ill patients” is a narrative review focusing on an important, and widely discussed problem. The scope of the review is too wide and the article promises to discuss “the barriers and facilitators to effectively treating critically ill patients at risk of nosocomial infection”.

Response: We have modified and followed the reviewer’s comments in order to improve the manuscript. We would like to thank the reviewer for his/her constructive criticism that has helped to make a better version of the paper.

Specific comments:

  1. The title does not properly describe the content of the article. The manuscript focuses on general information about antimicrobial resistance, resistance mechanisms, and even on microbiome.
  2. Bacteria names are misspelled in the whole document. For example, on Line 53 „Clostridium diffficile” is mentioned. The name changed in 2016, and the correct one is „Clostridioidesdifficile”

Response: As the sentence has been deleted, the bacteria has been deleted too.

  1. Line 53 – although the microbiome is an interesting topic, the line and later on, a whole section about microbiome does not fit in a narrative review titled „Practical lessons on antimicrobial therapy for critically ill patients”

Response: This has been reformulated as requested.

  1. “Colonisation” vs “Infection” section:the whole section focuses on colonization, while the infection is overlooked;

Response: We agree and the section name has been deleted and renamed as only colonization.

  1. „The role of the microbiome” section – does not fit in the scope of the review

Response: We agree and the entire section has been deleted.

  1. „Antimicrobial stewardship....” section – contains general, well-known information about the importance of antimicrobial stewardship

Response: As we are aiming for a pragmatical document, we have realised that this information is key for the intensivist, and as a practical lesson we had to start from here. We have, however, rewritten the entire part and expanded some parts; we also consider that AMS is not only de-escalation but also adequate to prescribe antibiotics. This is, in our opinion, very relevant and not always well addressed.

  1. „Which resistance mechanisms should an intensivist know” section – the title is catchy but the question was not answered in the text. The section describes a few resistance mechanisms. The information is not complete, and more importantly, it lacks novelty. Any clinician should have at least a basic understanding of the resistance mechanisms, especially if they have epidemiological value.

Response: We agree with these comments. We have included more information about why we should consider these mechanisms. We have included a paragraph that clarifies why we included these mechanisms. We also need to include a new paragraph to determine a better understanding of ICU settings for clinicians.

Overall, the review does not have a well-defined scope. The importance to the readership is poor, and the search strategy is not presented.  

Response: Thanks for these comments. We have taken all of these into consideration, and we have also added the search strategy after the introduction, as requested.

Reviewer 2 Report

Comments and Suggestions for Authors

The abstract can be further refined as it has some redundancy, especially in the first three sentences.

Specific comments:

In lines 70-71, there seems to be an extra word (fields)

Paragraph in lines 152-161 is very vague, not quite clear and is not well supported by evidence. 

Sentence "The primary aim of decreasing mortality is to tackle medical interventions using adequate tools." (lines 162-163) is not very informative. Suggest to reword or delete.

Figure 1 is not clear, and seems that not all fields are showing all text intended to be displayed. The source of information in the figure is not stated.

Sentence "AMR is a growing problem due to the need for the newest generations of antimicrobials." and its preceding statements in the same paragraph are redundant as the importance of AMR is already established in the introduction.

Figures 2 and 3 are not referred to in the text.

Furthermore, Figure 2 is used from a source that is not cited (either as source under the figure or in the references (https://www.nature.com/articles/s41581-019-0150-7))

In lines 230-232, no geographic scope - as the reference is global, it is good to refer to it as globally.

Future directions section is very scarce and does not always relate to the findings in the results section.

Comments on the Quality of English Language

Minor edits on uniformity: capitalisation of first letter for the same term (e.g. sepsis, vancomycin, etc.) should be uniformly used across the entire manuscript.

Author Response

The abstract can be further refined as it has some redundancy, especially in the first three sentences.

Response: The whole abstract has been rewritten and we hope it satisfies reviewer's comments: “Sepsis stands as a formidable global health challenge, with persistently elevated mortality rates in recent decades. Each year, sepsis not only contributes to heightened morbidity but also imposes substantial healthcare costs on survivors. In intensive care, targeted measures can be instituted to alleviate the incidence and impact of sepsis. These measures encompass the reduction of nosocomial infections and the prevention of equipment and patient colonisation by resilient pathogens. The overarching global crisis of bacterial resistance to newly developed antimicrobial agents intensifies the imperative for antimicrobial stewardship and de-escalation. This urgency has been accentuated in recent years, notably during the COVID-19 pandemic, as high-dose steroids and opportunistic infections presented escalating challenges. Ongoing research into airway colonization's role in influencing disease outcomes among critically ill patients underscores the importance of tailoring treatments to disease endotypes within heterogeneous populations. Looking ahead, the significance of novel antimicrobial delivery systems and drug monitoring is poised to increase. This narrative review delves into the multifaceted barriers and facilitators inherent in effectively treating critically ill patients vulnerable to nosocomial infections. The future trajectory of intensive care medicine hinges on the meticulous implementation of vigilant stewardship programs, robust infection control measures, and the continued exploration of innovative and efficient technological solutions within this demanding healthcare landscape.

Specific comments:

In lines 70-71, there seems to be an extra word (fields)

Response: we apologise for the type induced by the reference manager.

Paragraph in lines 152-161 is very vague, not quite clear and is not well supported by evidence. 

Response: Based on Reviewer’s comment, this paragraph has been deleted.

Sentence "The primary aim of decreasing mortality is to tackle medical interventions using adequate tools." (lines 162-163) is not very informative. Suggest to reword or delete.

Response:  The sentence has been rewritten to “ The central objective in reducing mortality is to address medical interventions employing appropriate tools. Laboratory testing can play a pivotal role in various aspects of AMS, particularly in facilitating de-escalation and optimizing the duration of interventions.

Figure 1 is not clear, and seems that not all fields are showing all text intended to be displayed. The source of information in the figure is not stated.

Response: The figure has been deleted.

Sentence "AMR is a growing problem due to the need for the newest generations of antimicrobials." and its preceding statements in the same paragraph are redundant as the importance of AMR is already established in the introduction.

Response: Thank you and the sentence has been deleted as clearly redundant.

Figures 2 and 3 are not referred to in the text.

Response: Both figures are now referred in the text.

Furthermore, Figure 2 is used from a source that is not cited (either as source under the figure or in the references (https://www.nature.com/articles/s41581-019-0150-7))

Response: Thanks for this. We have cited in the text and added the reference as suggested from Wang, T.Z., Kodiyanplakkal, R.P.L. & Calfee, D.P. Antimicrobial resistance in nephrology. Nat Rev Nephrol 15, 463–481 (2019). https://doi.org/10.1038/s41581-019-0150-7

In lines 230-232, no geographic scope - as the reference is global, it is good to refer to it as globally.

Response: We have added the word globally as suggested

Future directions section is very scarce and does not always relate to the findings in the results section.

Response: Thanks for these comments and we have expanded this part as suggested.

Minor edits on uniformity: capitalisation of the first letter for the same term (e.g. sepsis, vancomycin, etc.) should be uniformly used across the entire manuscript.

Response: We have revised the words again and hope no mistakes are found.

Reviewer 3 Report

Comments and Suggestions for Authors

The article addresses a current topic. For improvement, I recommend the following:
1. All abbreviations need to be checked; either they are used for the first time without definition, or they lack a definition, or they are defined upon second use.
2. The article requires a PRISMA flowchart and a Materials and Methods section.
3. In the chapter titled "The role of the microbiome," reference can also be made to the intestinal microbiome.
4. The figure 1 is difficult to understand.
5. Figure 3 - lacks a legend, and has no reference in the text.

Author Response

The article addresses a current topic. For improvement, I recommend the following:
1. All abbreviations need to be checked; either they are used for the first time without definition, or they lack a definition, or they are defined upon second use.

Response: We have checked and spelled out the abbreviations.

  1. The article requires a PRISMA flowchart and a Materials and Methods section.

Response: As this is a narrative review, we believe that a PRISMA flowchart does not apply. We have, however, included the search strategy immediately after the introduction.

  1. In the chapter titled "The role of the microbiome," reference can also be made to the intestinal microbiome.

Response: Based on reviewer 1, the section has been deleted

  1. The figure 1 is difficult to understand.

Response: The figure has been deleted based on this comment and reviewer 2.

  1. Figure 3 - lacks a legend, and has no reference in the text.

Response: Figure 3 complements the information in the text, and it has also been referenced in the text. Additionally, we have included that the figure was modified from the reference

Round 2

Reviewer 1 Report

Comments and Suggestions for Authors

The review title is “Practical lessons on antimicrobial therapy for critically ill patients”.

The newly written abstract focuses on the importance of measures that can be “instituted to alleviate the incidence and impact of sepsis” and it does not summarize the text. The abstract is more of an introduction.

The article has the following sections:

1.     Introduction – 5 paragraphs. The first one – it’s about antimicrobial therapy in ICU; the second one – is about increasing antimicrobial resistance; the third one – is about colonization as a predisposing factor to respiratory tract infections in critically ill patients. The section does not contain a clear aim. Without a stated aim, I will assume that the purpose of the review focuses on “practical lessons on antimicrobial therapy for critically ill patients”, as the title suggests. The search strategy should not be mentioned here but in a special section.

From my understanding, the article is now a “narrative review”. The search terms were “critical care”, “infection” and “antimicrobial therapy”. What are the practical lessons then? A clear scope is needed.

2.     The section “The role of colonization on critically ill patients” discusses the fact that colonization can be a predisposing factor for infection. This is well-known. Through what mechanisms? What are the practical lessons for ICU doctors? When should colonization be treated and when not? The topic is interesting. Lines 87-90 address a practical problem, but it is insufficiently described. How frequent is secondary Candida pneumonia? Why is it, not a good idea to treat Candida spp. (Line 89: “spp.” should not be italicized) isolated from sputum?

3.     The section “Antimicrobial stewardship and the dilemma of broad vs de-escalation in ICU” – an interesting topic (and in alignment with the title) but why is „de-escalation” a dilemma? It seems like a good option and it is a well-established important step of AMS. Are there arguments for using broad-spectrum antibiotics when a more targeted therapy is available?

4.     Resistance mechanisms

Line 159-260: „It is essential to understand the mechanisms of resistance when choosing alternative therapy”. Why? Explaining why would be a „practical lesson”.

The overall quality of the manuscript improved, but the novelty and the clarity of the presented information are still questionable. The article needs a clear scope. After establishing the scope of the article, the structure will follow naturally, as the scope dictates the information that belongs in the article.

The abstract should be a brief summary of the text, presenting the most important information for the reader.

Comments on the Quality of English Language

Minor editing of English language required

Author Response

The review title is “Practical lessons on antimicrobial therapy for critically ill patients”.

The newly written abstract focuses on the importance of measures that can be “instituted to alleviate the incidence and impact of sepsis” and it does not summarize the text. The abstract is more of an introduction.

Many thanks for this observation, the abstract has been altered to reflect a summary of the text.

The article has the following sections:

  1. Introduction – 5 paragraphs. The first one – it’s about antimicrobial therapy in ICU; the second one – is about increasing antimicrobial resistance; the third one – is about colonization as a predisposing factor to respiratory tract infections in critically ill patients. The section does not contain a clear aim. Without a stated aim, I will assume that the purpose of the review focuses on “practical lessons on antimicrobial therapy for critically ill patients”, as the title suggests. The search strategy should not be mentioned here but in a special section.

From my understanding, the article is now a “narrative review”. The search terms were “critical care”, “infection” and “antimicrobial therapy”. What are the practical lessons then? A clear scope is needed.

Thank you for this suggestion. The aim of the article has been clearly laid out and the methods section added to a separate section. The scope has been more clearly laid out.

  1. The section “The role of colonization on critically ill patients” discusses the fact that colonization can be a predisposing factor for infection. This is well-known. Through what mechanisms? What are the practical lessons for ICU doctors? When should colonization be treated and when not? The topic is interesting. Lines 87-90 address a practical problem, but it is insufficiently described. How frequent is secondary Candidapneumonia? Why is it, not a good idea to treat Candida spp. (Line 89: “spp.” should not be italicized) isolated from sputum?

Thank you for this feedback, we agree this is an important issue and have added a more descriptive section on secondary Candida pneumonia.

  1. The section “Antimicrobial stewardship and the dilemma of broad vs de-escalation in ICU” – an interesting topic (and in alignment with the title) but why is „de-escalation” a dilemma? It seems like a good option and it is a well-established important step of AMS. Are there arguments for using broad-spectrum antibiotics when a more targeted therapy is available?

Thank you for this advice, we feel that de-escalation in very sick patients is unadvisable and have added an explanation here.

  1. Resistance mechanisms

Line 159-260: „It is essential to understand the mechanisms of resistance when choosing alternative therapy”. Why? Explaining why would be a „practical lesson”.

 Many thanks, an explanation has been added here.

The overall quality of the manuscript improved, but the novelty and the clarity of the presented information are still questionable. The article needs a clear scope. After establishing the scope of the article, the structure will follow naturally, as the scope dictates the information that belongs in the article.

The abstract should be a brief summary of the text, presenting the most important information for the reader.

Many thanks for this observation. The abstract has been altered to reflect a summary of the text.

Reviewer 3 Report

Comments and Suggestions for Authors

The article has been significantly improved. Figure 2 requires a legend.

Author Response

The article has been significantly improved. Figure 2 requires a legend.

Figure legend is now included.